# BRIGHTDREAMER: GENERIC 3D GAUSSIAN GENERATIVE FRAMEWORK FOR FAST TEXT-TO-3D SYNTHESIS

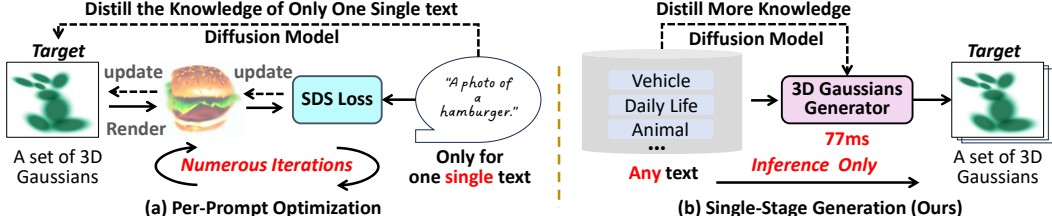

Figure 1: **A comparison between per-prompt optimization-based methods, and our single-stage generation-based approach with an end-to-end objective.** **(a)** Optimization-based methods directly initialize a 3D representation model, *e.g.*3D Gaussian Splatting (GS). This process usually suffers from slow per-sample optimization (*e.g.*, several hours for a single text). **(b)** By contrast, once trained, our approach directly generates 3D content for any unseen text prompt in **77 ms with a single run of a feed-forward of our generator**.

## ABSTRACT

Text-to-3D synthesis has recently seen intriguing advances by combining the text-to-image models with 3D representation methods, *e.g.*, Gaussian Splatting (GS), via Score Distillation Sampling (SDS). However, a hurdle of existing methods is the low efficiency, per-prompt optimization for a single 3D object. Therefore, it is imperative for a paradigm shift from per-prompt optimization to one-stage generation for any unseen text prompts, which yet remains challenging. A hurdle is *how to directly generate a set of millions of 3D Gaussians to represent a 3D object*. This paper presents ***BrightDreamer***, an end-to-end single-stage approach that can achieve generalizable and fast (**77 ms**) text-to-3D generation. Our key idea is to **formulate the generation process as estimating the 3D deformation from an anchor shape with predefined positions**. For this, we first propose a Text-guided Shape Deformation (**TSD**) network to predict the deformed shape and its new positions, used as the centers (one attribute) of 3D Gaussians. To estimate the other four attributes (*i.e.*, scaling, rotation, opacity, and SH coefficient), we then design a novel Text-guided Triplane Generator (**TTG**) to generate a triplane representation for a 3D object. The center of each Gaussian enables us to transform the triplane feature into the four attributes. The generated 3D Gaussians can be finally rendered at **705 frames per second**. Extensive experiments demonstrate the superiority of our method over existing methods. Also, BrightDreamer possesses a strong semantic understanding capability even for complex text prompts. *The project code is available at* `http://`.

## 1 INTRODUCTION

Text-to-3D generation has recently received considerable attention in the computer graphics and vision community owing to its immersive potential across diverse applications, such as virtual reality and gaming (Li et al., 2023a).

Recently, with the emergence of diffusion models (Ho et al., 2020; Rombach et al., 2022) and neural rendering techniques (Mildenhall et al., 2021; Kerbl et al., 2023), text-to-3D has witnessed an unprecedented technical advancement. In particular, pioneering methods, such as DreamFusion (Poole et al., 2022), LatentNeRF (Metzer et al., 2023), SJC (Wang et al., 2023a), have sparked significant

Table 1: The comparison of different training paradigms of 3D generative model. "Interpolation" can reflect the capability of detail control and the continuity of the learned space.

| Method | Generation Time | Training Data | Interpolation | Text Input | Representative |
|---|---|---|---|---|---|
| GAN-Based | less than 1 second | 2D Images | ✓ | ✗ | GRAF, Pi-GAN, StyleNeRF |
| Per-Prompt | several hours | ✗ | ✗ | ✓ | DreamFusion, LucidDreamer |
| 3D Diffusion | several seconds | 3D data | ✗ | ✓ | Latte3D |
| Single-Stage | less than 1 second | ✗ | ✓ | ✓ | Instant3D (NeRF, lower text richness) **Ours** (**3D GS, higher text richness**) |

interest in the research community, catalyzing a trend toward developing techniques for creating 3D assets from texts. The follow-up methods then focus on either quality improvement (Raj et al., 2023; Shi et al., 2023; Wang et al., 2023b; Liang et al., 2023) or geometry refinement (Chen et al., 2023a; Lin et al., 2023) or training efficiency (Tang et al., 2023; Yi et al., 2023).

The dominant paradigm of these methods is to randomly initialize a 3D representation model, *e.g.*, Neural Radiance Fields (NeRF) (Mildenhall et al., 2021) or Gaussian Splatting (Kerbl et al., 2023), and optimize such a model to align with a specific text prompt, as depicted in Fig. 1 (a). Unfortunately, these methods suffer from two critical challenges. ***Firstly***, as per-prompt optimization usually requires several tens of thousands of iterations, this inefficiency brings a considerable obstacle to broader applications. It is significantly different from the mainstream training paradigm in the field of 2D image generation (Song et al., 2020; Rombach et al., 2022) or 3D-aware image generation (Schwarz et al., 2020; Chan et al., 2021; Jiang et al., 2023; Or-El et al., 2022; Chan et al., 2022): *a generative model is trained with a collection of text-image pairs or images, and the model can generate the desired content from any input at the inference stage*. We show the main difference of the different paradigms in 3D generative models in Tab. 1. ***Secondly***, as demonstrated in Fig. 2(a), existing methods often fail to accurately process the complex texts. For example, the mainstream methods all fail to generate 3D content that input prompt contains complex interaction between multiple entities. This limitation arises from the models being trained on a single text prompt, which results in a degraded capability in comprehensive semantic understanding.

Therefore, it is urgently needed for a paradigm shift from per-prompt optimization to develop a *generic* text-to-3D generation framework. Once trained, the generative framework should be able to generate content from any text prompts in the inference stage, as depicted in Fig. 1(b). Furthermore, given the scarcity of 3D data in comparison to the abundance of 2D image data, leveraging well-trained 2D image diffusion models to facilitate the training of 3D generative models presents a more effective and resource-efficient approach. Previously, some research efforts, *e.g.*, ATT3D (Lorraine et al., 2023) and Instant3D (Li et al., 2023b), have been made grounded in NeRF representation. The core insight is to add a large number of texts and take them as conditional inputs to generate explicit spatial representations, such as triplane (Chan et al., 2022). Nonetheless, in stark contrast to the volume rendering in NeRF, 3D GS representation for an object usually consists of millions of 3D Gaussians. Consequently, there exists an inherent and natural difficulty in converting the generation representations into 3D GS ones in their framework.

In this paper, we propose ***BrightDreamer***, an *end-to-end single-stage* framework that, for the **first** time, can achieve generalizable and fast (77 ms) text-to-3D GS generation. BrightDreamer exhibits a robust ability for complex semantic understanding (Fig. 2 (a)), and it demonstrates a substantial capacity for generalization (Fig. 2 (b)). In addition, same as traditional generative models (Goodfellow et al., 2014), our generator can interpolate between two inputs (Fig. 2 (c)), which enables users to fully engage their imagination and creativity, expanding the potential for novel and nuanced design exploration. As stated before, the 3D GS representation of an object usually consists of **several millions** of 3D Gaussians. Consequently, directly generating such an extensive collection, is impractical. **Our key idea** is to redefine this generation problem as its equal problem, *i.e.*, 3D shape deformation. Specifically, we place and fix some anchor positions to form the initial shape. Then, it can be deformed to the desired shape by giving different input prompts through our designed Text-guided Shape Deformation (**TSD**) network (Sec. 3.1). After the deformation, the new positions can be set to the centers of the 3D Gaussian. Upon establishing the basic shape, we elaborately design a Text-guided Triplane Generator (**TTG**) to generate a spatial representation of the 3D object (Sec. 3.2). Subsequently, we utilize the spatial feature of each center of 3D Gaussian to represent its whole feature and translate it into the remaining attributes (including scaling, rotation, opacity, and SH coefficient) through our well-designed Gaussian Decoder (Sec. 3.3). For TTG, grounded on our re-analysis of the previous convolution-based triplane generation process, we have identified and solved two primary deficiencies that necessitate rectification. One issue pertains to the spatial inhomogeneity observed during the calculation process, as illustrated in Fig. 4. The other issue arises

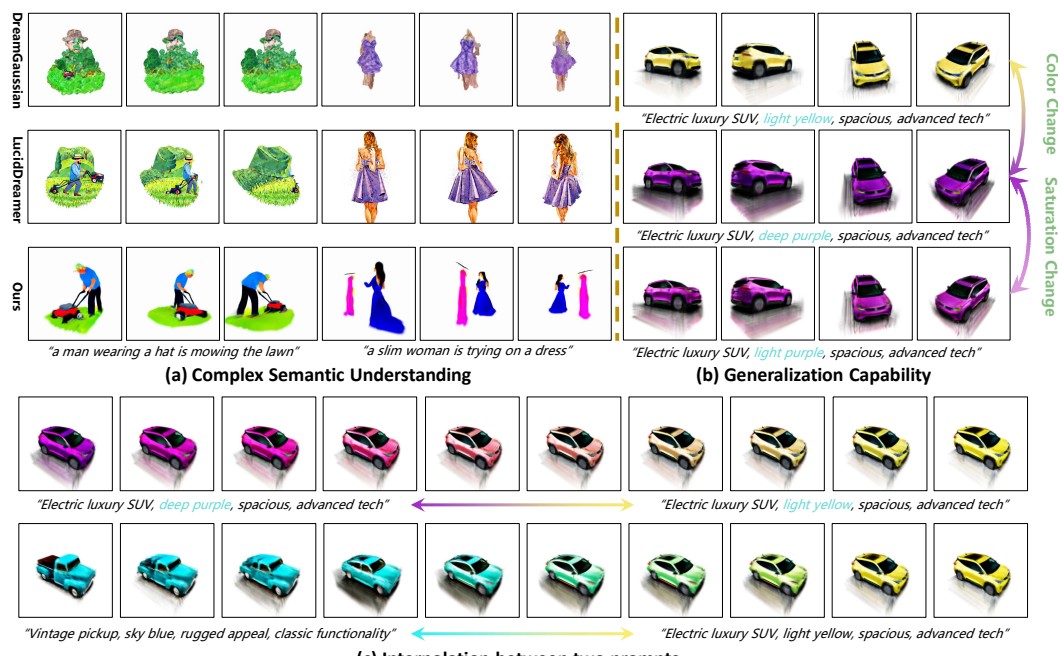

Figure 2: DreamGaussian (Tang et al., 2023) and LucidDreamer (Liang et al., 2023) are both optimized for a single text. Our result is the direct generation. And for the display of our generalization, **all the prompts do not appear in our training set**. (a) is for showing the complex text understanding. (b) is for demonstrating our capability of generalization. It is noteworthy that **light purple**, **deep purple**, and **light yellow** don't appear in the training set. (c) Interpolation between two prompts from color and shape perspective.

from the single-vector style control mechanism similar to StyleGAN (Karras et al., 2019), which complicates the management of relationships between multiple entities.

Our contributions can be summarized as follows: (**I**) We propose BrightDreamer, the first 3D Gaussian generative framework to achieve generalizable and fast text-to-3D synthesis. (**II**) We design the Text-guided Shape Deformation (TSD) network to simplify the difficulty of direct generation of 3D Gaussians. We design the Text-guided Triplane Generator (TTG) to generate the object's spatial features and then decode them as the 3D Gaussians. For TTG design, we re-analyze and solve the existing problems in the mainstream triplane generator, including spatial inhomogeneity and text understanding problems. (**III**) Extensive experiments demonstrate that BrightDreamer not only can understand the complex semantics (while the per-prompt optimization methods fail) but also can utilize its generalization capability to achieve generation control.

## 2 RELATED WORKS

**Text-to-3D Generation.** Existing methods can be grouped into two categories. **1) Optimization-based methods** typically commence with a randomly initialized 3D model, such as NeRF (Mildenhall et al., 2021), and subsequently employ text-image priors (Radford et al., 2021; Rombach et al., 2022) to guide and optimize its parameters. After undergoing thousands of iterative refinements, this predefined 3D model progressively morphs to embody the shape described by the corresponding text input. DreamField (Jain et al., 2022) represents the inaugural foray into text-to-3D methodology, utilizing the pre-trained text-image model, CLIP (Radford et al., 2021), as a guiding mechanism for the optimization process of a predefined NeRF model. DreamFusion (Poole et al., 2022) proposes the Score Distillation Sampling (SDS) to transfer the prior of the 2D diffusion model (Ho et al., 2020) into a 3D representation model (Mildenhall et al., 2021; Müller et al., 2022), which achieves impressive performance and ignites the research enthusiastic for the text-to-3D task. VSD (Wang et al., 2023b) and ISM (Liang et al., 2023) are devoted to re-designing the SDS loss (Poole et al., 2022), enabling much more local details of the 3D model. MVDream (Shi et al., 2023) and Perp-Neg (Armandpour et al., 2023) attempt to solve the *Janus* problem, *i.e.*, multi-face problem in some text prompts. **2) Generation-based methods**, by contrast, aim to directly generate a 3D model from

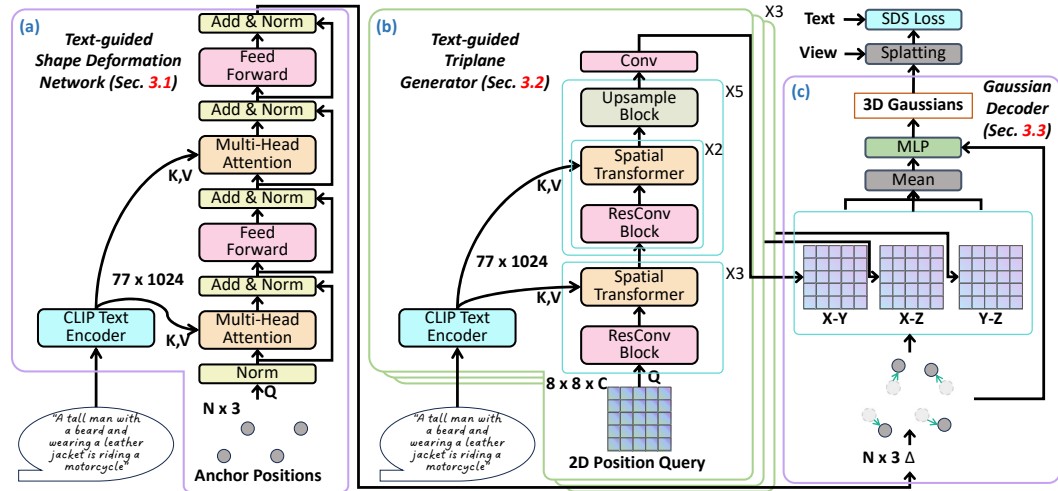

Figure 3: **An overview of our BrightDreamer**. The details of Spatial Transformer, ResConv Block and Upsample Block are shown in Fig. 5.

a given text, streamlining the process of text-to-3D generation. ATT3D (Lorraine et al., 2023) is the first attempt to train a NeRF model with multiple texts. Instant3D (Li et al., 2023b) designs some modules to map the text input to the EG3D model (Chan et al., 2022) and then use SDS to train this model. Nonetheless, different from the volume rendering in NeRF, 3D GS representation for an object usually consists of millions of 3D Gaussians. *Therefore, it is inherently difficult to convert the NeRF representations into 3D GS ones*. We propose BrightDreamer, a generic framework that, for the ***first*** time, can achieve fast (77 ms) text-to-3D GS generation. BrightDreamer exhibits a robust ability for complex semantic understanding (see Fig. 2).

**3D Gaussian Splatting (GS).** Recently, 3D GS (Kerbl et al., 2023) has become the most popular 3D representation method of 3D objects or scenes. 3D GS shows a faster rendering speed and higher application potential than NeRF (Mildenhall et al., 2021). Within a short time, a large number of methods have been proposed to leverage 3D GS for diverse tasks, *e.g.*, anti-aliasing novel view synthesis (Yu et al., 2023; Yan et al., 2023b), SLAM (Yan et al., 2023a; Keetha et al., 2023; Matsuki et al., 2023; Yugay et al., 2023), human reconstruction (Li et al., 2024; Moreau et al., 2023; Kocabas et al., 2023; Abdal et al., 2023; Li et al., 2023c; Liu et al., 2023), dynamic scene reconstruction (Luiten et al., 2023; Yang et al., 2023b; Wu et al., 2023; Yang et al., 2023a; Xu et al., 2023), and 3D content generation (Xu et al., 2024; Chen et al., 2023b; Tang et al., 2023; Yi et al., 2023; Li et al., 2023d; Liang et al., 2023). Our work is also based on 3D GS; however, we aim to develop a generic text-to-3D generation framework that can generate 3D Gaussians at a low latency (77 ms).

## 3 METHOD

**Overview.** The objective of BrightDreamer is to directly generate 3D Gaussians in response to text prompts. After training, BrightDreamer is capable of generating the 3D Gaussians with a remarkably low generation latency (about 77ms). And, the generated 3D Gaussians can be rendered at an impressive inference speed of over 700 frames per second. 3D Gaussians can be defined by five attributes, namely the center $p'$, scaling $S$, rotation $R$, opacity $\alpha$, and SH coefficient $SH$. To directly generate 3D Gaussians, **our key idea** is two-fold: 1) Defining *anchor positions*, *i.e.*, predefined positions, to estimate the center of 3D Gaussians; 2) Building *implicit spatial representation*, which can be decomposed to estimate the other four attributes of 3D Gaussians.

Intuitively, we propose BrightDreamer, and an overview is depicted in Fig. 3. Given a text prompt as input, we transform it to a $77 \times 1024$ embedding through the frozen CLIP text encoder. Next, the TSD network (Sec. 3.1) transforms the fixed anchor positions to the desired shape with text guidance. The new positions are used as the centers of 3D Gaussians. We then design the TTG (Sec. 3.2) to separately generate three feature planes to construct the implicit spatial representation. Based on the centers of Gaussians, we can obtain their spatial features, which are then transferred to the other attributes through the Gaussian Decoder (Sec. 3.3). Finally, we render 3D Gaussians to

2D images and use the SDS Loss (Poole et al., 2022) to optimize the whole framework. We now describe our BrightDreamer in detail.

## 3.1 TEXT-GUIDED SHAPE DEFORMATION (TSD)

**The goal of TSD** is to *obtain the center (one attribute) of each 3D Gaussian*. Considering that directly outputting a huge number of center coordinates is extremely difficult, we overcome this hurdle by deforming the anchor positions instead of generating them.

**Anchor Position.** The *anchor position* is predefined in this paper, which is a fixed coordinate. It serves as one of the inputs for the TSD. Specifically, we place the anchor positions on the vertices of a 3D grid, as represented by the gray points in Fig. 3. Then, we design the TSD network to predict their deviation to deform the initialized shape of the 3D grid, guided by the text prompt input.

**Network Design.** As shown in Fig. 3 (a), the inputs of TSD are text prompts and anchor positions. Firstly, the text prompts are encoded as the text embedding by an off-the-shelf text encoder, *e.g.*, CLIP (Radford et al., 2021) or T5 (Raffel et al., 2020). Considering the possibility of the complex input sentence, it remains non-trivial how to bridge each position and word in the sentence. The cross-attention (Vaswani et al., 2017) can quantify the correlation degree between each point and each word within a sentence. We then employ the cross-attention to design a module to obtain the deviation from the anchor position. It consists of the Layer Normalization (Ba et al., 2016), Multi-Head Attention, Feed-Forward Network, and shortcut connection (Vaswani et al., 2017). Consequently, certain positions, correlating more closely with corresponding words in the sentence, are assigned with higher attention scores. This process enables the aggregation of features that more accurately reflect the characteristics of the corresponding words. The detailed computation process is formulated as follows:

$$output = FFN(softmax(\frac{W_Q(p)W_K(y)}{\sqrt{d}}) \cdot W_V(y)), \tag{1}$$

where $p \in \mathbb{R}^3$ is the 3D coordinate of the anchor position, $y \in \mathbb{R}^{77 \times 1024}$ is the text embedding of the input prompt, 77 and 1024 are the sentence length and the embedding dimension, $W_Q(\cdot)$, $W_K(\cdot)$ and $W_V(\cdot)$ are the *query*, *key*, *value* transformation function, $d$ is the feature dimension, *score* represent the attention score between words and points, $h$ is the intermediate feature, $FFN(\cdot)$ is the feed-forward network. The output of the TSD network is the offset $\Delta \in \mathbb{R}^3$ of the anchor positions. To ensure the stability of the training, we control the maximum extent to which a point can deviate from the anchor position. Specifically, given the degree of freedom $\beta \in \mathbb{R}$, we use the following equation to adjust the range of *output* into interval $(-\beta, \beta)$:

$$\Delta = 2 \cdot \beta \cdot sigmoid(output) - \beta. \tag{2}$$

Finally, we have the deformed position $p' \in \mathbb{R}^3$, *i.e.* centers of 3D Gaussian, corresponding to input prompt, formulated as follows:

$$p' = p + \Delta. \tag{3}$$

## 3.2 TEXT-GUIDED TRIPLANE GENERATOR (TTG)

Upon determining the centers of the 3D Gaussians, we need to obtain the other four attributes. To efficiently assign features to each Gaussian, **the objective of TTG** is to *generate an implicit spatial representation in space*, represented by the triplane. Therefore, we design a novel and highly efficient triplane generator, whose input is also the text prompts.

**One challenge** is that the previous triplane generation approaches, such as EG3D (Chan et al., 2022) and Instant3D (Li et al., 2023b), exhibit the problem of spatial inhomogeneity, as shown in Fig. 4. Since they directly segment a feature map into three feature maps along the channel dimension, only a few areas are computed together. For example, the position (0, 0) in the 2D space is unfolded to (0, 0, :), (0, :, 0), and (:, 0, 0), denoted by blue color in Fig. 4. Taking $1 \times 1$ *Conv* as an example, only these three areas are calculated together. On the contrary, (0, 0, :) is hardly possible to be calculated with (0, :, 1), because they do not appear at the same pixel in the 2D feature map. The same applies to the $3 \times 3$ *Conv*. This means that only a few areas share the same spatial information, while the others do not, thus causing spatial inhomogeneity. *For this, a simple yet effective way is to apply three generators (without sharing weights).*

**Another challenge** is that given the complex prompts, squeezing a sentence into a single style feature vector to apply AdaIN (Karras et al., 2019; 2020; 2021) could result in a loss of local details. Therefore, we need a more fine-grained generation method guided by the word level, thus can retain more information of text encoder trained on large-scale dataset. *Naturally, calculating cross-attention between the pixels of the feature map and words in the sentence is a better choice.*

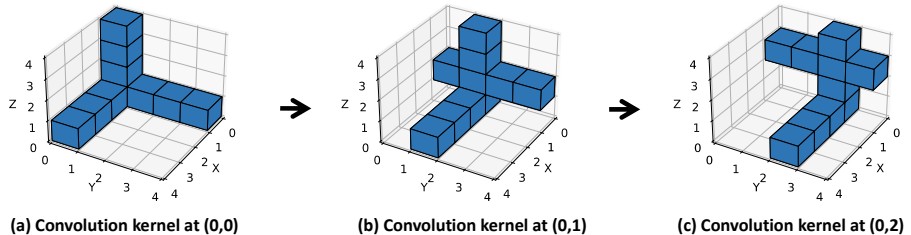

**(a) Convolution kernel at (0,0)**   **(b) Convolution kernel at (0,1)**   **(c) Convolution kernel at (0,2)**

Figure 4: **The visualization of expanding 2D convolution kernel (cyan area) to 3D and its moving process in previous convolutional triplane generator (Chan et al., 2022)**. We use $1 \times 1$ convolutional kernel as an example. Only several positions are interacted, which leads to spatial inhomogeneous.

To address these two challenges, we design the Text-guided Triplane Generator (TTG), as shown in Fig. 3 (b). Our TTG is designed with the inspiration from the spatial transformer block and residual convolutional block in Stable Diffusion (Rombach et al., 2022). Considering the increased computational demand associated with pixel-wise self-attention in the feature map of (Rombach et al., 2022), we do not incorporate this layer into our network. Instead, we find that interleaved convolutional layers can sufficiently facilitate the interaction within the feature map. The detailed designs are shown in Fig. 5. For the whole pipeline, we first initialize a 2D input query according to its 2D trigonometric function position encoding. Three ResConv Blocks and Spatial Transformer blocks are stacked to assemble the prompt word features at low resolution through cross-attention. We then gradually increase the resolution of the feature map through the stacks of ResConv Blocks, Spatial Transformer blocks, and Upsample Block by five times, as depicted in Fig. 3 (b). Finally, we use a *Conv* layer to output the plane feature. We describe the design of the Spatial Transformer block, ResConv block, and Upsample block in detail.

**Spatial Transformer Block.** As Fig. 5 (a) shows, the Spatial Transformer Block comprises two multi-head cross-attention modules and a feed-forward network (Vaswani et al., 2017). The process is initiated by flattening the 2D feature map into a 1D structure, thereby transforming the dimensions from $(H, W, C)$ to $(H \times W, C)$, with each pixel's feature considered as the input query embedding. Subsequent to this transformation, the features undergo normalization via Layer Normalization (Ba et al., 2016). The normalized features serve as queries, while the text embeddings act as keys and values in the computation of the cross-attention feature. This cross-modality attention mechanism is designed to align the feature map with the corresponding words in the input sentence. Following the application of two cross-attention modules, the features are further refined through a feed-forward network. This sequence of operations also incorporates the use of skip connections, mirroring the original transformer architecture (Vaswani et al., 2017), to facilitate effective feature processing and integration.

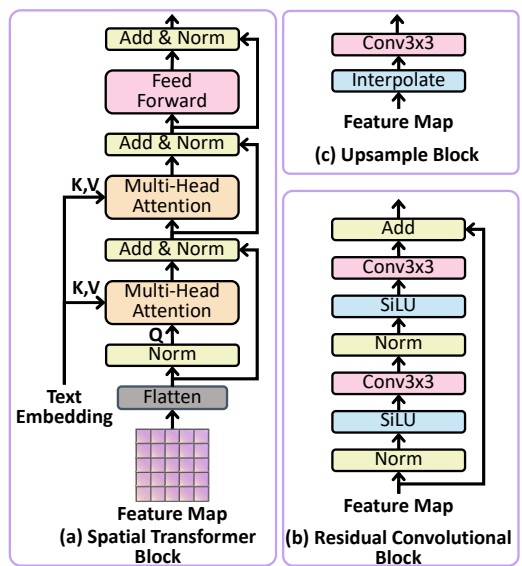

Figure 5: **A detailed illustration of specific blocks.** (a) Spatial Transformer Block. (b) Residual Convolutional Block. (c) Upsample Bock.

**ResConv Block.** As illustrated in Fig. 5 (b), the Residual Convolutional Block consists of a sequence of components: a Layer Normalization layer, followed by the application of the SiLU activation function (Elfwing et al., 2018), and $3 \times 3$ convolutional layers. Then a skip connection is established between input and output.

**Upsample Block.** As depicted in Fig. 5 (c), the Upsample Block begins with the interpolation of the feature map, enlarging it by a factor of $2\times$. Following this upscaling, the enlarged feature map is further processed through a $3 \times 3$ convolutional layer.

### 3.3 3D GAUSSIANS DECODER

It aims to obtain the other four attributes of 3D Gaussian for achieving the generation. Upon generating the triplane, comprised of the three feature planes $\pi_{xy}$, $\pi_{xz}$, $\pi_{yz}$, we can obtain the feature vector $\mathcal{F} \in \mathbb{R}^{32}$ of each Gaussian based on its center $p'$. Subsequently, the feature vector needs to be converted into additional attributes of 3D Gaussian, including *opacity* $\alpha \in \mathbb{R}$, *scaling* $S \in \mathbb{R}^3$, *rotation* $R \in \mathbb{R}^4$, and *SH coefficient*.

Specifically, we first project the 3D coordinate onto the three planes, X-Y, X-Z, and Y-Z. Based on the projected 2D coordinates, we can derive the features $\mathcal{F}_{xy}$, $\mathcal{F}_{xz}$, $\mathcal{F}_{yz}$ according to the interpolation with their four vertex in the 2D feature maps. To ensure that the gradient back-propagation is evenly distributed across all three planes, we utilize an averaging operation to aggregate these features, thereby obtaining the 3D Gaussian's feature $\mathcal{F}$. Given that the attributes of 3D Gaussian can be categorized into two groups, *i.e.*, shape and color, we develop two distinct transformation modules $F_{shape}$ and $F_{color}$. Each module is a lightweight, two-layer Multi-Layer Perceptron (MLP) network. To enhance the gradient back-propagation to the TSD, the center of 3D Gaussian $p'$ is additionally inputted into both modules:

$$\alpha, S, R = F_{shape}(\mathcal{F}, p'), \ SH = F_{color}(\mathcal{F}, p'). \tag{4}$$

In the training process, we observe the scaling $S$ is not a stable variation, and the memory consumption of 3D Gaussian rendering is extremely sensitive to it. Therefore, we use the following equation to control it to interval $(a, b)$:

$$S = (b - a) \cdot sigmoid(S) + a. \tag{5}$$

Upon obtaining all attributes of 3D Gaussian, our generation process is completed. We can render it from arbitrary view direction to 2D images.

### 3.4 OPTIMIZATION

Our training commences with the selection of $B$ prompts from the training set. These prompts are then fed into our 3D Gaussians generator, which is tasked with generating the corresponding 3D GS representation of the objects. Following this, we proceed to randomly sample $C$ view directions to render $C$ 2D images. The $B \times C$ rendered images are supervised through the Score Distillation Sampling (SDS) loss function (Poole et al., 2022), as Eq. 6 shows, in conjunction with the Perp-Neg (Armandpour et al., 2023). In this way, our generator can gradually construct a mapping relationship between text and 3D.

$$\nabla_\theta \mathcal{L}_{\text{SDS}}(\phi, \mathbf{x} = g_\theta(prompt)) \triangleq \mathbb{E}_{t,\epsilon} \left[ w(t) \left( \hat{\epsilon}_\phi \left( \mathbf{z}_t; y, t \right) - \epsilon \right) \frac{\partial \mathbf{x}}{\partial \theta} \right], \tag{6}$$

where $prompt$ is the input prompt of the generator, $\theta$ denotes the trainable parameters of 3D Gaussians generator, $\phi$ denotes the parameters of denoising network, $x$ is the generated image by process $g_\theta(\cdot)$, $w(t)$ is the weighting function with time step $t$ in the denoising schedule, $\epsilon$ is the random noise, $z_t = x + \epsilon$, $y$ is the adjusted text according to the sampling view direction, $\hat{\epsilon}_\phi(\cdot)$ is the predicted noise.

## 4 EXPERIMENTS

### 4.1 IMPLEMENTATION DETAILS

All our experiments are conducted on a server with 8 GPUs with 80GB memory. Our codebase is constructed on the PyTorch framework (Paszke et al., 2019) with Automatic Mixed Precision Training (AMP). And we introduce the Gradient Checkpointing technology (Chen et al., 2016) to save the GPU memory. We use the Adam optimizer (Kingma & Ba, 2014) to update the parameters of our generator with a constant learning rate of $5 \times 10^{-5}$, $\beta_1$ of 0.9 and $\beta_2$ of 0.99. We train our generator using the images rendered at $512 \times 512$ resolution by original rasterization of 3D Gaussian Splatting (Kerbl et al., 2023) and use the DeepFloyd IF (Stablility, 2023) UNet to calculate the SDS Loss (Eq. 6). The prompt batch size is set to 64 in total and the camera batch size is set to 4. We set the freedom $\beta$ (Eq. 2) to 0.2, and the range of scaling $(a, b)$ (Eq. 5) to $(-9, -3)$. The anchor position is placed as a $64^3$ 3D grid, and the resolution of the generated triplane is $256 \times 256$. We use the following three sets of prompts to train a single model.

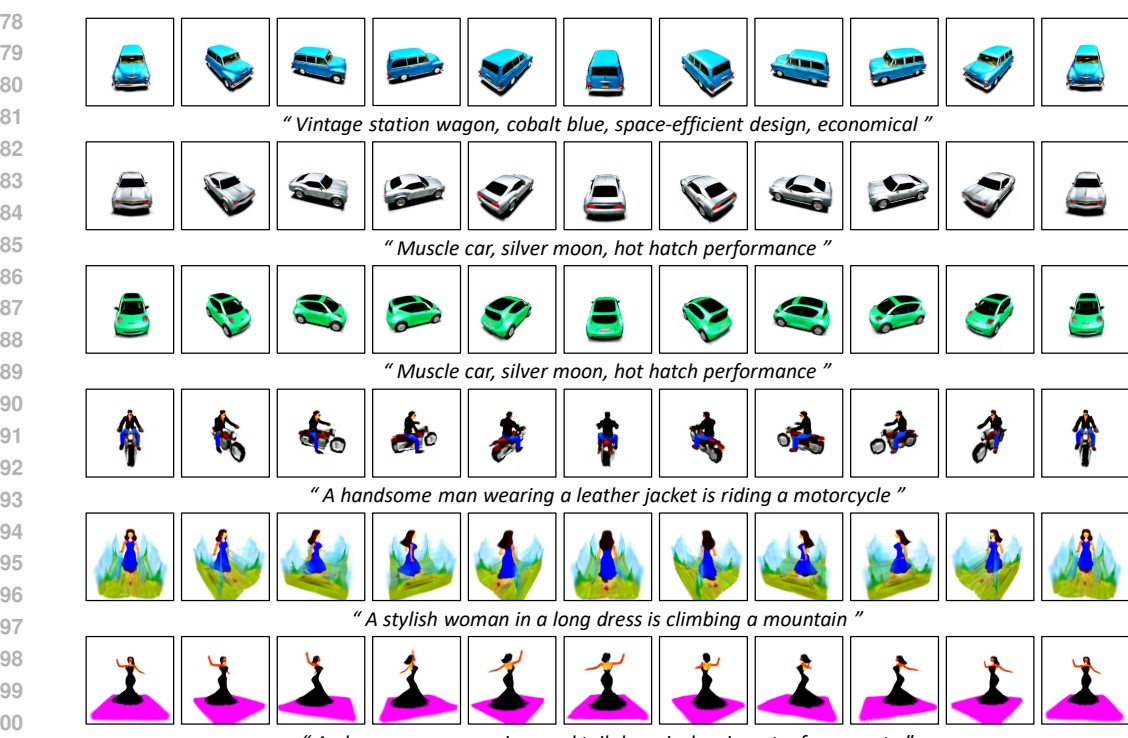

Figure 6: **Generation Demonstration.** All prompts do not appear in the training set.

## 4.2 DATASETS

**Vehicles.** We construct a vehicle's prompts set using ChatGPT[1] to generate a description about car details, *e.g.*, `"vintage convertible, cherry red, chrome bumpers, white-wall tires"`. There are 3,026 prompts containing 1,856 unique words in total.

**Daily Life.** This dataset is constructed by Instant3D (Li et al., 2023b). They use the ChatGPT[1] to generate more than $17,000$ prompts containing $3,135$ unique words.

**Animals.** This prompt set is introduced by Instant3D (Li et al., 2023b). And the structure is `"a {species} sitting {item} and wearing {gadget} and wearing a {hat}"`. There are $3,150$ prompts in total.

## 4.3 DEMONSTRATION OF BRIGHTDREAMER

To show the quality and view consistency, we show some results in Fig. 6. All prompts don't appear in the training set. We also show much more results in *supplementary material*.

**Generalizability.** Furthermore, we find that **even some unseen words** can also be understood correctly. For example, the word "banana" is not in our training set, but our BrightDreamer can also understand its color. We discuss this phenomenon in Sec. D.

**Inference Latency and Rendering Speed.** In Tab. 2, we show the inference latency on a single A800 GPU and a single RTX3090 GPU. BrightDreamer can generate a 3D GS representation for any text prompt in **less than 0.1 seconds**, showing a large margin improvement, compared to optimization-based methods, which need **several hours** to optimize for a single prompt. The generated 3D Gaussians can be rendered at over **700 FPS speed**.

Table 2: The generation latency (millisecond) and rendering speed (FPS, Frames Per Second).

| Device | Generation Latency | Rendering Speed |
|---|---|---|
| RTX 3090 24GB | 79 ms | 698 FPS |
| A800 80GB | 77 ms | 705 FPS |

## 4.4 COMPARISON WITH OTHER METHODS

For effective evaluation, our experiment setting is that *none of the test prompts are present in the training set while those of the per-prompt optimization methods (Poole et al., 2022; Wang et al.,*

---

[1]https://chat.openai.com/

[2]https://github.com/threestudio-project/threestudio

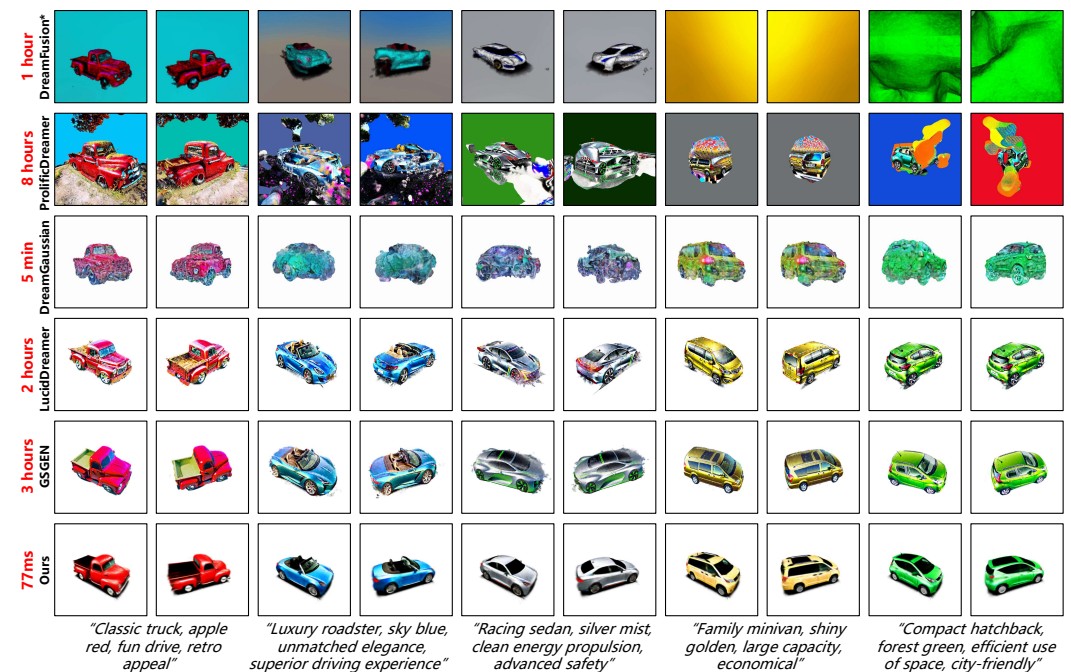

Figure 7: **Vehicle generation comparison.** All prompts don't appear in the training set. The result of Dreamfusion is reproduced by ThreeStudio[2].

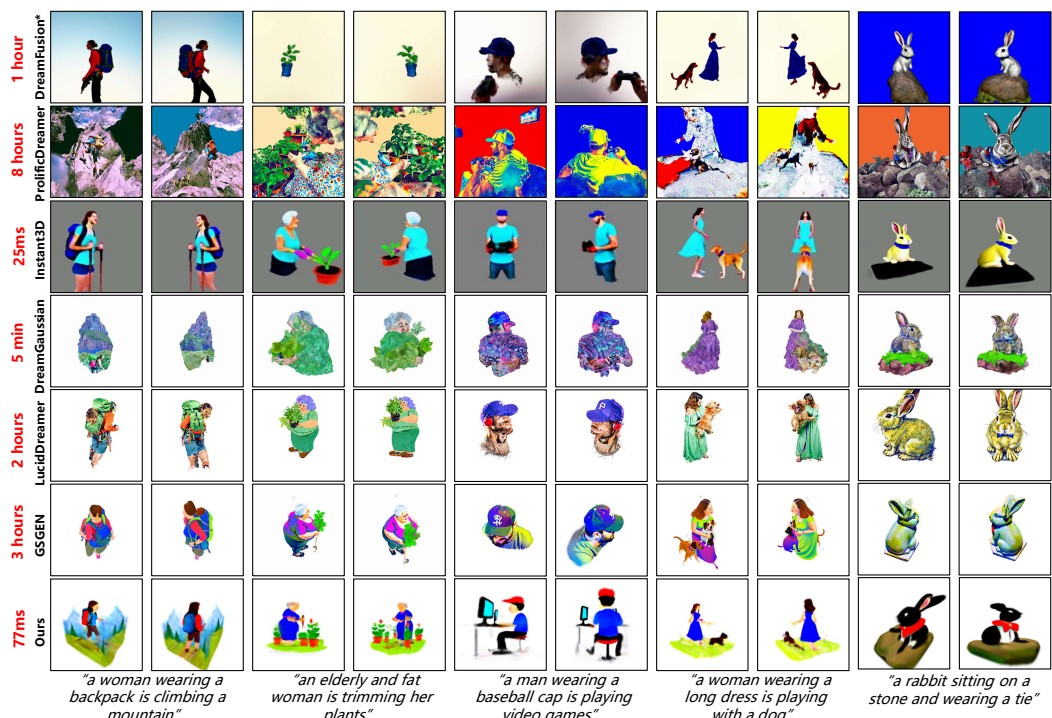

Figure 8: **Comparison for daily life and animal.** All prompts don't appear in the training set.

*2023b; Tang et al., 2023; Liang et al., 2023; Chen et al., 2023b) are not.* This setting highlights the generalization capability of our one-stage generation.

**Qualitative comparison.** In Fig. 7, we show the comparison results. Note that other methods need training different models for different prompts. In contrast, our method can directly infer one-time for a single prompt and all the test prompts do not appear in our training prompt set. Due to the car generation limitations remaining in Stable Diffusion, we use DeepFloydIF denoising UNet

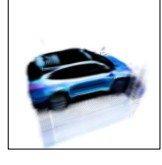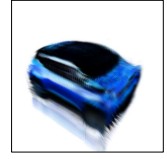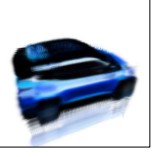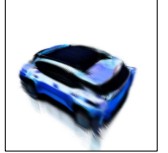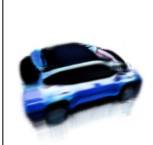

(a) Complete Design  (b) w/o Triplane Generator Division  (c) w/o Coordinate Shortcut

Figure 9: **The key components in our design.** All the models are trained to 10, 000 iterations with the same configuration. (a) Our completion design. (b) Replace the three separate generators with a single generator. (c) Don't input the coordinate to $F_{shape}$ and $F_{color}$.

in Eq. 6. However, it can only provide the $64 \times 64$ supervision, so there remains some unfairness. We first compare with SoTA NeRF optimization methods, dreamfusion (Poole et al., 2022) and ProlificDreamer (Wang et al., 2023b). Their generation result is extremely unreasonable and chaotic, which shows the difficulty when working on relatively complex prompts. Though Dream-Gaussian's training is relatively fast, 5 minutes, the generation result is still relatively bad. Since LucidDreamer (Liang et al., 2023) and GSGEN (Chen et al., 2023b) introduce the Point-E (Nichol et al., 2022) as their prior or supervision, they can generate the correct shape of a vehicle. However, LucidDreamer has abnormal light spots, which cause unreal sensing. Our method can deal with arbitrary text prompts in **77 ms**, which shows an extremely strong application value. Importantly, you can scale up to check the second two columns in Fig. 7. Looking through the window at the seats inside, there is an irregular phenomenon in the seat of GSGEN and LucidDreamer. But ours looks reasonable, which shows that joint training can make the network have a stronger capability of semantic understanding.

In Fig. 8, we compare our method to Instant3D (Li et al., 2023b) and other per-prompt optimization methods based on daily life and animal datasets. Though ProlificDreamer (Wang et al., 2023b) and LucidDreamer (Liang et al., 2023) improve the original SDS loss, which allows them can get more details in the generated content, the consistency to the prompt is damaged a lot.

**Quantitative Comparison.** As shown in Tab. 3, we provide the percentage numerical comparison of human preference choice. We present

Table 3: Human Preference.

| DreamFusion | ProlificDreamer | Instant3D | LucidDreamer | **Ours** |
|---|---|---|---|---|
| 4.8% | 16.2% | 29.5% | 5.7% | **43.8%** |

participants with five rendered videos along with the corresponding text prompts generated by five baseline models, allowing them to select their preferred option for each case. Most of the users expressed a preference for the content generated by our model.

### 4.5 ABLATION STUDIES

In Fig. 9, we demonstrate the influence of our network design for training. Compared between Fig. 9 (a) and Fig. 9 (b), our divided triplane generator can reduce the degree of chaos in the space significantly, which shows the necessity of our division. As Fig. 9 (a) and Fig. 9 (c) demonstrate, it is necessary to pass the coordinate into the $F_{shape}$ and $F_{color}$. This design can construct a gradient pathway toward the TSD Network, ensuring more accurate shape formulation.

## 5 CONCLUSION

In this paper, we introduced the first text-driven 3D Gaussians generative framework, Bright-Dreamer, capable of generating 3D Gaussians within a remarkably low latency of 77ms. To address the challenge of directly generating a vast quantity of Gaussians, millions of them, we innovatively deform anchor positions and use the new positions as the centers of 3D Gaussians, in response to the input prompt. This approach effectively circumvented the obstacle of generating a large number of positions. Regarding network architecture, we thoroughly reevaluated the triplane generation process and introduced an improved alternative strategy. Our largest contribution is poised to significantly advance the field of generalized 3D generation, offering a novel and efficient pathway to creating 3D assets from text prompts immediately. Extensive experiments prove that our Bright-Dreamer has a strong complex semantic understanding capability and strong generalization ability.

**Future work.** First, the spatial resolution of the generated 3D model is relatively low, resulting in a lack of fine-grained detail. Second, the range of supported scene types could be further expanded to enhance the model's versatility. To achieve these two targets, integrating higher-resolution diffusion models and expanding the dataset with additional text prompts are promising research directions.

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
