# OpenReview forum: "BrightDreamer: Generic 3D Gaussian Generative Framework for Fast Text-to-3D Synthesis"
_ICLR.cc/2025/Conference — ICLR 2025 Conference Withdrawn Submission_

### Official Review · Reviewer_UAaa · 2024-10-29

**Soundness:** 2
**Presentation:** 2
**Contribution:** 2
**Rating:** 3
**Confidence:** 2

**Summary:**

This paper introduces a novel amortized Text-to-3D Gaussian generation method, enabling the creation of 3D Gaussian structures from text prompts in a single forward pass. The proposed approach incorporates two key innovations:

- Anchor-based Deformation:
Traditional 3D Gaussian representations require numerous Gaussian points to model complex scenes, making direct generation challenging. To address this, the authors introduce a strategy where a small number of anchor points are fixed and subsequently deformed through a text-conditioned transformer. These deformed points act as the central points for generating the 3D Gaussian structures.
- Text-guided Triplane Generator (TTG):
The paper further proposes the use of a Text-guided Triplane Generator (TTG) to create a triplane structure. The anchor points are then utilized to query Gaussian features from the generated triplane.

To train the models, the Score Distillation Sampling (SDS) method is applied to rendered images, following the practice used in previous amortized Text-to-3D models. This ensures effective training of the networks. The authors conducted extensive experiments, demonstrating the effectiveness of their approach compared to traditional train-from-scratch Text-to-3D Gaussian methods.

**Strengths:**

This paper consist of following strengths:

- **Introduction of 3D Gaussians to Amortized Text-to-3D Generation:**

This paper is the first to apply 3D Gaussians to the amortized Text-to-3D generation task, demonstrating higher efficiency compared to traditional methods that require training from scratch.

- **Reframing the 3D Gaussian Generation Problem:**

The paper transforms the challenge of generating 3D Gaussians into a deformation problem. This approach addresses the difficulty of generating complex objects that would typically require millions of Gaussian points, making direct generation infeasible.

- **Proposal of a Triplane Generator for Spatial Features:**

A novel triplane generator is introduced to produce spatial features, which can be decoded into Gaussian attributes. The method is further refined with several specific design improvements to enhance performance.

- **Experimental Validation of Efficiency:**

Extensive experiments validate the efficiency of the proposed approach compared to original Text-to-3D Gaussian generation methods.

**Weaknesses:**

Though this method achieves promising results through its proposed designs, the paper presents several weaknesses that need to be addressed:

**1.Novelty:**

While the paper is the first to reformulate 3D Gaussian generation as a deformation problem, this concept is not new in the field of explicit 3D generation. Similar deformation-based approaches have been explored in previous works, such as:

- Wang, Nanyang, et al. "Pixel2mesh: Generating 3d mesh models from single rgb images." Proceedings of the European conference on computer vision (ECCV). 2018.
- Wen, Chao, et al. "Pixel2mesh++: Multi-view 3d mesh generation via deformation." Proceedings of the IEEE/CVF international conference on computer vision. 2019.

Since 3D Gaussians represent a novel but explicit 3D format, the deformation approach is straightforward to apply, suggesting that this work is more of an incremental adaptation than a fundamentally novel idea.

It is accessable to bring an old idea to a novel field. However, another critical issue lies in the representation ability. While Gaussian splatting typically involves millions of points to accurately capture scene details, this paper proposes using a fixed, smaller number of anchor points. The lack of discussion on how many anchor points are required for convincing results raises concerns about the expressiveness of the generated models. If the number is reduced too much, the method may struggle to represent complex scenes effectively. The authors should address this limitation by providing a more thorough analysis and justification for their approach.

**2.Network Structure:**

The network architecture closely resembles the one proposed by Zou et al. in:

- Zou, Zi-Xin, et al. "Triplane meets gaussian splatting: Fast and generalizable single-view 3d reconstruction with transformers." Proceedings of the IEEE/CVF CVPR. 2024.

Both networks feature a triplane decoder and a point-based querier. Although the authors claim to introduce improvements, the paper does not provide an ablation study to evaluate the impact of these modifications. A more in-depth discussion and comprehensive ablation studies are necessary to demonstrate the value of these improvements.

**3.Experiments:**

The experimental results focus only on comparisons with text-to-3D Gaussian generation methods. However, since the paper's task aligns more closely with amortized Text-to-3D generation, it should also be compared against those methods to ensure a fair evaluation.

Additionally, as mentioned earlier, an ablation study is needed to assess the effectiveness of the network design choices. This would clarify the role of individual components and provide insights into which modifications contribute to the improved performance.

**4.Writing:**

The paper introduces new terms, such as "Triplane Generator Division" and "Coordinate Shortcut," in ablation study, but fails to explain them clearly. As a result, it becomes difficult to understand the exact meaning and purpose of these concepts. Providing detailed descriptions of these definitions would improve clarity and make the paper more accessible to readers.

**Questions:**

As noted in the Weaknesses section, I believe the authors should clarify the motivation behind their approach and the network structure similarty with triplane meet gaussians. Therefore, I am giving a rating of 3. If the authors address these points, I would be open to raising my score.

---

### Official Review · Reviewer_DLaw · 2024-11-02

**Soundness:** 2
**Presentation:** 2
**Contribution:** 2
**Rating:** 5
**Confidence:** 5

**Summary:**

The paper presents a generic text-to-3D generation method. Instead of performing score distillation to optimize a single 3D representation one at a time, the proposed method aims for a single-stage solution, where a text prompt is used to generate a set of deformations from predefined anchored points as well as three feature planes. The generated deformed points are then used to query the feature planes to construct 3D Gaussians, where the positions of the Gaussians are at the deformed points themselves, and the remaining four attributes of the Gaussians (scale, rotation, opacity, spherical harmonic coefficients) are established from the feature planes. The results show that the proposed method can generate 3D objects from text prompts in just ~77 ms.

**Strengths:**

The paper addresses an important problem in text-to-3D generation, where the current speed to generate a 3D object using optimization-based methods are relatively low, taking minutes on average to generate an object. By making the generation process entirely feed forward, the method directly samples a 3D object in milliseconds.

The paper is quite well written. I appreciate Table 1 which clearly demonstrates the differences of the method with previous work.

**Weaknesses:**

1) The technical content suffers from two notable issues.

- The training seems to be done for each class of object, e.g., vehicle, animal, etc. For per-prompt optimization methods, the benefit is that no such class definition is required. It remains unclear whether why such categorization is required, or minimum how many samples per class is required for training. Additionally, some data must be prepared for training.

- While the inference speed is at milliseconds, there seems to have some fidelity-speed tradeoff. The generated 3D model quality is not as good as per-prompt optimization case.

From the above points, my guess is that training a generative framework like the proposed method is actually a trade-off. It leads to some limitations in the generative process (limited to object categories) as well as some degradations in object quality. This makes it tricky to handle arbitrary prompts like in optimization case.


2) The experiment results appear to be quite limited. Some results are not particularly convincing.

- The qualitative results appear to be more blurry than optimization-based text-to-3D Gaussian methods.
     The 3D models also do not have a lot of details, and the colors appear to be very saturated.

- The generalization of the model (to unseen prompts) are not well demonstrated. In terms of diversity and unseen objects it seems the generalization is worse than an image diffusion model as the current training scale is way smaller.

- Fig. 9: I do not see clear improvement between a, b, c.

- The training is quite expensive, with 30+ hours on 8 x 80GB GPUs.


3) The paper writing can be further improved by addressing the following issues:

- Some newer works for per-prompt optimization methods can be cited. Currently some 2024 methods are missing.
  [A] Taming Mode Collapse in Score Distillation for Text-to-3D Generation, CVPR 2024
  [B] DiverseDream: Diverse Text-to-3D Synthesis with Augmented Text Embedding, ECCV 2024

- The discussion of network blocks in 3.2 is quite long. Some of the details are not directly relevant to 3D generation and can be moved to appendix (Spatial Transformer Block, ResConv, UpSample).

- Reduce the use of bold, italic text in the writing. Too much emphasis diluted the emphasis.

- Some paragraphs are particularly dense. Some spaces between paragraphs should be reserved.

**Questions:**

1. Could the authors provide a comment on the quality-speed trade-off? How can this issue be potentially addressed?

2. Could the authors comment on the diversity and the generalization of the model?

3. Could the authors explain the choice of training data? How are the current training prompts selected? What happens if the prompts are just randomly sampled?

**Details Of Ethics Concerns:**

All good.

---

### Official Review · Reviewer_GZMt · 2024-11-03

**Soundness:** 2
**Presentation:** 2
**Contribution:** 1
**Rating:** 3
**Confidence:** 4

**Summary:**

This paper introduces a real-time text-to-3D synthesis method called BrightDreamer, where a Text-guided Shape Deformation (TSD) network and a Text-guided Triplane Generator (TTG) are proposed to predict the attributes of a 3DGS representation.

**Strengths:**

1. This paper proposes a real-time text-to-3D synthesis method based on 3DGS, which does not require 3D data for training.
2. The proposed Text-guided Shape Deformation (TSD) and Text-guided Triplane Generator (TTG) are novel and provide a certain level of technical contribution.

**Weaknesses:**

1. The quality of the generated content by the proposed method is not high enough. The generated texture looks over-smoothed and lacks realism, e.g., the rabbit in Figure 8. The generated geometry is incomplete, e.g., the car in Figure 9(a).
2. The comparison experiments are not fair enough. (a) The methods compared in Section 4.4 are insufficient. Some of the latest 3D diffusion methods should also be included in the comparison, e.g., Latte3D [1] and LGM [2]. (b) The author needs to ensure the quality of the generated content. For instance, although ProlificDreamer may have issues with multiple facets, the generated content in Figures 7 and 8 is clearly below its typical standard.
3. The summary of existing methods in Table 1 is not comprehensive. For example, there are GAN-based generators that also support text input [3-9].

[1] LATTE3D: Large-scale Amortized Text-To-Enhanced3D Synthesis. [2] LGM: Large Multi-View Gaussian Model for High-Resolution 3D Content Creation. [3] Text2shape: Generating shapes from natural language by learning joint embeddings. [4] CLIP-Forge: Towards Zero-Shot Text-to-Shape Generation [5] Shapecrafter: A recursive text-conditioned 3d shape generation model. [6] Towards implicit text-guided 3d shape generation. [7] Autosdf: Shape priors for 3d completion, reconstruction and generation. [8] CLIP-Sculptor: Zero-Shot Generation of High-Fidelity and Diverse Shapes from Natural Language [9] Hierarchical Text-Conditional Image Generation with CLIP Latents.

**Questions:**

See the weaknesses section.

---

### Official Review · Reviewer_cs7N · 2024-11-03

**Soundness:** 2
**Presentation:** 3
**Contribution:** 2
**Rating:** 5
**Confidence:** 4

**Summary:**

The paper presents an amortized text-to-3D Gaussian generator trained with SDS loss. The framework consists of two modules: TSD, responsible for center deformation, and TTG, which generates other Gaussian attributes. The model achieves fast 3D Gaussian generation.

**Strengths:**

1.	The model achieves fast 3D Gaussian generation, requiring only 77 ms.
2.	The paper finds that re-parameterizing the Gaussian’s scaling helps stabilize optimization.
3.	The architecture and model design are reasonable.

**Weaknesses:**

1.	In Table 1, the timing for Latte3D should be less than 1 second. Additionally, the method is not ‘3D diffusion’ but a model trained with amortization. ATT3D is missing from Table 1.
2.	For the baseline comparison in Figures 7 and 8, it would be better to include SDS with the MVdream prior, which is much more robust and produce higher-quality shapes.
3.	The experimental results are not fully convincing. Specifically: 1) the baseline comparison does not include the improved version of SDS methods with MBdream; 2) in Figure 9, only one pair of results is presented to demonstrate the advantage of the complete design. For this ablation study, running an FID or CLIP score evaluation on a holdout set would better illustrate the complete design’s effectiveness compared to other settings.

**Questions:**

1.	How many iterations are needed to train this model?
2.	What are the parameter counts for the TSD and TTG modules, respectively?

---

### Note · Authors · 2024-11-13

I have read and agree with the venue's withdrawal policy on behalf of myself and my co-authors.